# High-Yield Growth and Tunable Morphology of Bi_2_Se_3_ Nanoribbons Synthesized on Thermally Dewetted Au

**DOI:** 10.3390/nano11082020

**Published:** 2021-08-07

**Authors:** Raitis Sondors, Gunta Kunakova, Liga Jasulaneca, Jana Andzane, Edijs Kauranens, Mikhael Bechelany, Donats Erts

**Affiliations:** 1Institute of Chemical Physics, University of Latvia, 19 Raina Blvd., LV-1586 Riga, Latvia; raitis.sondors@lu.lv (R.S.); gunta.kunakova@lu.lv (G.K.); liga.jasulaneca@lu.lv (L.J.); jana.andzane@lu.lv (J.A.); edijs.kauranens@lu.lv (E.K.); 2Institut Européen des Membranes, IEM-UMR 5635, ENSCM, CNRS, University of Montpellier, Place Eugène Bataillon, 34095 Montpellier, France; mikhael.bechelany@umontpellier.fr; 3Faculty of Chemistry, University of Latvia, 19 Raina Blvd., LV-1586 Riga, Latvia

**Keywords:** Bi_2_Se_3_, nanoribbon, synthesis, physical vapor deposition

## Abstract

The yield and morphology (length, width, thickness) of stoichiometric Bi_2_Se_3_ nanoribbons grown by physical vapor deposition is studied as a function of the diameters and areal number density of the Au catalyst nanoparticles of mean diameters 8–150 nm formed by dewetting Au layers of thicknesses 1.5–16 nm. The highest yield of the Bi_2_Se_3_ nanoribbons is reached when synthesized on dewetted 3 nm thick Au layer (mean diameter of Au nanoparticles ~10 nm) and exceeds the nanoribbon yield obtained in catalyst-free synthesis by almost 50 times. The mean lengths and thicknesses of the Bi_2_Se_3_ nanoribbons are directly proportional to the mean diameters of Au catalyst nanoparticles. In contrast, the mean widths of the Bi_2_Se_3_ nanoribbons do not show a direct correlation with the Au nanoparticle size as they depend on the contribution ratio of two main growth mechanisms—catalyst-free and vapor–liquid–solid deposition. The Bi_2_Se_3_ nanoribbon growth mechanisms in relation to the Au catalyst nanoparticle size and areal number density are discussed. Determined charge transport characteristics confirm the high quality of the synthesized Bi_2_Se_3_ nanoribbons, which, together with the high yield and tunable morphology, makes these suitable for application in a variety of nanoscale devices.

## 1. Introduction

Bismuth Selenide (Bi_2_Se_3_) is a layered narrow bandgap semiconductor, which has been widely studied and demonstrated potential for application in optical recording systems [1], photochemical devices [2], battery electrodes [3], and thermoelectric devices [4]. This material belongs to the recently discovered class of three-dimensional topological insulators (TI), exhibiting conducting states with nondegenerate spins protected by time-reversal symmetry [5]. The simple surface states make Bi_2_Se_3_ an ideal candidate to realize various unique physical phenomena, such as quantum anomalous Hall effect, superconductivity, topological magnetoelectric effect, enhancement of thermoelectric figure of merit [5,6,7,8,9,10], as well as the bulk bandgap of Bi_2_Se_3_ specifies great potential for possible high-temperature spintronic applications [11].

However, the unique properties of Bi_2_Se_3_ surface states are challenging to utilize practically, as the bulk carriers overwhelm the surface carriers. Bi_2_Se_3_ nanostructures, such as nanoplates, nanowires, and nanoribbons, have emerged as perfect candidates for exploiting properties of the topological surface states due to their high surface-to-volume ratio [12,13]. For example, complete suppression of bulk conduction has been demonstrated in Bi_2_Se_3_ nanoribbons thinner than 30 nm [13]. The surface-to-volume ratio has also been shown to play an important role in tuning thermoelectric properties [14]. Bi_2_Se_3_ nanoribbons can reach lengths of tens of micrometers and even millimeters [15], allowing one to easily make multiple electrical connections to probe topological transport properties, as well as can serve as field-effect transistor channels [16] and active elements in nanoelectromechanical devices [17,18]. Successful applications of Bi_2_Se_3_ nanostructures as nanowires and nanoribbons have been demonstrated in the areas of thermoelectrics [19,20,21], photodetection [22], topological insulator devices [23,24,25], and nanoelectromechanical devices [17,18].

Both applications of Bi_2_Se_3_ nanoribbons in nanoelectromechanical devices and fabrication of electron transport devices employing topological insulator properties of the material require certain morphology of the nanoribbons. Methods for obtaining Bi_2_Se_3_ nanostructures include sonochemical [26], solvothermal [27], metal–organic chemical vapor deposition [28], and both catalyst-free and catalyst-assisted physical vapor deposition [12,27]. Catalyst-free physical vapor deposition methods are especially attractive due to the high quality of the synthesized nanostructures and contamination-free process [29]; however, these methods result in the formation of diverse nanostructures as nanoplates, nanoribbons, and nanowires of random morphology formed during one synthesis run. For practical applications in devices, it is necessary to control the morphology during the synthesis and to increase the yield of the synthesized nanostructures with the required geometry.

A good candidate for control of the morphology of the Bi_2_Se_3_ nanoribbons is catalyst-assisted synthesis, which has been widely exploited for the production of highly crystalline nanoribbons that exhibit distinct topological properties [30]. In this method, either as-deposited or pre-annealed (dewetted) Au thin film or colloidal Au particles have been used as catalysts promoting vapor–liquid–solid (VLS) [30] or vapor–solid–solid (VSS) [31] growth of the Bi_2_Se_3_ nanostructures. In the classical vapor–liquid–solid (VLS) growth reported for the 20–30 nm Au nanoparticles, the source material in the vapor phase diffuses into a liquid-metal catalyst. The concentration of the source material is increased until the solubility limit is reached and precipitation of the solid nanostructure takes place. Depending on the size and temperature-dependent surface and body diffusion coefficients of adatoms, the nucleation and further growth of the nanostructures can occur on the top (the catalyst nanoparticle remains on the substrate, the nanostructure grows up) or bottom (the catalyst nanoparticle remains on the tip of the grown nanostructure) surface of the catalyst nanoparticle [32]. In the VSS growth mode, the vapor of the source material penetrates the catalyst particle remaining in the solid state; however, there are very limited reports on Au-catalyzed VSS growth of the Bi_2_Se_3_ nanostructures claiming the occurrence of this mode instead of VLS when the Au nanoparticle size is 300 nm and above [31]. While it has been reported that the presence of the Au catalyst influences the Bi_2_Se_3_ nanostructure morphology evolution at the initial growth stage, most of the studies of VLS Bi_2_Se_3_ growth employ only one particular size of the catalyst nanoparticles or thickness of the catalyst layer. To the best of knowledge, there are no reports on the systematic investigation of the dependence of morphology and yield of the Bi_2_Se_3_ nanoribbons on the size and areal number density of the Au catalyst nanoparticles.

In this work, the Bi_2_Se_3_ nanoribbon morphology and yield are studied in relation to the diameter and areal number density of Au nanoparticles formed during the pre-annealing of Au catalyst layers of different thicknesses. The contribution of two different growth mechanisms—catalyst-free and vapor–liquid–solid—to the outcome of the Bi_2_Se_3_ nanoribbon synthesis and mechanical transfer of the nanoribbons to other substrates resulting in the selective transfer of nanoribbons of certain geometry is discussed.

Charge transport properties of the Bi_2_Se_3_ nanoribbons grown on the Au catalyst layer promoting the highest yield indicate consistency with the previously reported nanoribbon magnetotransport characteristics.

## 2. Materials and Methods

Au films with thicknesses from 1.5 to 16 nm were deposited on microscope glass slides (25 × 75 mm^2^) using thermal evaporation in a Sidrabe SAF EM metal deposition system. The films were annealed in an inert atmosphere by heating from room temperature to 585 °C for 45 min at a rate of 13 °C/min at a starting pressure of 2 Torr. The temperature was kept at 585 °C for 15 min, after which the substrate was allowed to cool naturally. Then, the size distribution and the areal number density of the formed Au nanoparticles were characterized using a scanning electron microscope (SEM, Hitachi FE-SEM S-4800, Tokyo, Japan). The obtained SEM images were analyzed with ImageJ software.

Bi_2_Se_3_ nanoribbons were grown on the as-produced dewetted Au substrates [29]. The source material (99.999% Bi_2_Se_3_ powder, Sigma-Aldrich, Burlington, MA, USA) was placed in the center of the quartz furnace tube where the temperature reaches a maximum of 585 °C during the deposition process. The substrates were placed downstream from the source material where the temperature reaches a maximum of 375–490 °C during the deposition process. The furnace was flushed with nitrogen for 3–5 min, then heated from room temperature at a rate of 13 °C/min for 45 min at a starting pressure of 2 Torr until the temperature in the center of the furnace reached 585 °C. The furnace was held at 585 °C for 15 min and cooled down to 535 °C at a cooling rate of 8 °C/min. When the temperature had dropped to 535 °C, a nitrogen gas flow at a constant pressure of 25 Torr was introduced into the furnace tube to induce the growth of nanoribbons. When the temperature had dropped to 475 °C, the synthesis was terminated by filling the tube with nitrogen until the pressure had reached 1 atm.

Number density and distribution of lengths and widths of the nanoribbons were determined by SEM by inspection of as-grown samples. The thicknesses of the nanoribbons were determined using an atomic force microscope (AFM, Asylum Research MFP-3D, Santa Barbara, CA, USA). For AFM characterization, the as-synthesized nanostructures were mechanically transferred to Si/SiO_2_ chips by pressing the chip against the region of interest on the substrate.

The crystalline structure of the synthesized samples was studied using X-ray diffraction spectroscopy (powder diffractometer X’PERT MRD with Cu Kα radiation source, Malvern Panalytical Ltd., Malvern, UK), ref. card No. 96-901-1966 [33].

For determination of transport properties, nanoribbons grown on a dewetted Au layer of initial thickness 3 nm were transferred to Si/SiO_2_ substrates, and electron beam lithography (Raith, eLine) followed by metal evaporation (5 nm Ti/80 nm Au, Sidrabe, Riga, Latvia) was used to fabricate electrical contacts. Magnetotransport measurements were performed with a Quantum Design DynaCool 9T physical property measurement system (San Diego, CA, USA). Resistance was recorded in a four-probe measurement setup, and the magnetic field was applied perpendicular to the nanoribbon top surface.

## 3. Results and Discussion

### 3.1. Yield and Morphology

Au nanoparticles with mean diameter from 10 nm to 148 nm and number density from 9 nps/µm^2^ to 8000 nps/µm^2^ were formed by dewetting Au layers with initial thickness from 1.5 nm to 16 nm, with thicker initial Au layers yielding larger Au nanoparticles (Figure 1a–c).

The effect of the initial Au layer thickness h on the mean dewetted Au nanoparticle diameter 〈D〉 was described, approximating
〈D〉 by a power function of h as $\langle D\rangle \;∼h^{Z}$ [19] (Figure 2a). The blue line shows the equation fitted to the experimental data. Approximating the nanoparticles as perfect spheres and assuming conservation of volume, the resultant Au nanoparticle number density N can be estimated by dividing the initial Au volume per substrate area $\frac{V_{L}}{A_{S}}$ by the mean volume of the nanoparticle $V_{NP}$:$$N=\frac{V_{L}}{A_{S}\cdot V_{NP}}=\frac{6h}{\pi {\langle D\rangle }^{3}}.$$

By comparing experimentally measured and calculated Au nanoparticle number density (Figure 2b), it can be seen that the theoretical model describes the experimental data reasonably well.

SEM images of the substrate with deposited Bi_2_Se_3_ showed that the surface of the substrate was covered with a layer of Bi_2_Se_3_ nanostructures (Figure 1d–f). Energy-dispersive X-ray spectroscopy measurements confirmed stoichiometric nanostructure growth in a region where the maximum temperature reached 443 ± 9 °C during Bi_2_Se_3_ deposition. In total, more than 70,000 measurements were made in this region to determine nanowire Bi_2_Se_3_ nanoribbon geometrical parameters.

The areal number density of the Bi_2_Se_3_ nanoribbons vs. the initial Au layer thickness showed pronounced non-linear dependence with a maximum at 3 nm thin Au layer (Figure 3a); however, the number density of the nanoribbons per 1000 nanoparticles showed a linear increase with the increase of the nanoparticle diameter (Figure 3b), which may indicate increasing domination of the VLS growth mechanism over the catalyst-free growth, as discussed in detail in Section 3.2 of this article. The mean length of the synthesized Bi_2_Se_3_ nanoribbons showed almost no changes for the initial Au layer thicknesses below 5 nm and correspondingly for the Au nanoparticles with diameters below 50 nm; however, the increase of the initial thickness of the Au layer above 5 nm and consequently of the sizes of Au nanoparticle above 50 nm resulted in the pronounced increase of the mean lengths of the Bi_2_Se_3_ nanoribbons (Figure 3c,d). Bi_2_Se_3_ nanoribbon length dependence indicates that larger Au nanoparticles correspond to the increased growth rate of the nanoribbons, possibly because of an increased Au catalyst surface area, which enables a greater Bi_2_Se_3_ throughput. Positive dependences on the initial thickness of the Au layer and on the diameter of the Au nanoparticles were also observed for the mean thickness of the Bi_2_Se_3_ nanoribbons (Figure 3e,f). The nearly linear relationship between Bi_2_Se_3_ nanoribbon thickness and Au nanoparticle size can be explained by the effect of surface stresses at the VLS triple-phase junction during nanoribbon growth [34]—the angle between the Au nanoparticle and Bi_2_Se_3_ nanoribbon α is a function of surface tensions: $sin\alpha =\gamma_{VS}/\gamma_{VL}=T/D_{NP}$, where $\gamma_{VS}$ is the vapor/solid surface tension, $\gamma_{VL}$ is the vapor/liquid surface tension, T is nanoribbon thickness and $D_{NP}$ is nanoparticle diameter. If the angle between the Au nanoparticle and Bi_2_Se_3_ nanoribbon surfaces is constant, then the nanoribbon thickness is expected to be directly proportional to nanoparticle diameter. In contrast, the mean width of the Bi_2_Se_3_ nanoribbons vs. Au layer thickness as well as vs. Au nanoparticle diameter showed pronounced non-linear dependence with a minimum at 3 nm thin Au film and Au nanoparticles with diameters ~10 nm (Figure 3g,h). Presumably, the widths of the nanoribbons may depend on the prevailing nucleation and growth mode and be governed either by the size of the Au nanoparticle or by the nanoplate seed, as is discussed in detail in the next section. It should also be noted that the widths of the nanoribbons were extracted from the SEM images of as-grown samples by measuring the visualized projections of the nanoribbons, which does not allow high-accuracy determination of actual nanoribbon width.

### 3.2. Growth Mechanism

A significant part of the nanoribbons was found to be growing from the nanoplates seeds (Figure 4a), which is consistent with the growth mechanism observed in the similar two-step catalyst-free synthesis process [13,29] when the width of the nanoribbon is governed by the width of the nanoplate facet it starts from.

The XRD spectra of Bi_2_Se_3_ nanostructures synthesized on the dewetted layers of Au particles of mean diameters ~10 and ~85 nm, obtained from 3 nm and 13 nm thick Au layers, respectively, with the XRD spectra of Bi_2_Se_3_ nanostructures grown on a glass substrate by the catalyst-free method showed a decreased intensity of diffraction peaks corresponding to Bi_2_Se_3_ nanostructure growth with the c-axis perpendicular to the substrate surface (peaks at 9.26°, 18.58°, and 27.94° correspond to Bi_2_Se_3_ crystallographic planes (003), (006), and (009), all belonging to the (003n) group), accompanied by an increase of intensity for diffraction peaks related to other Bi_2_Se_3_ crystallographic planes (104), (015), (1010), and (110) (Figure 5). According to previous reports, the increase of intensity of these XRD peaks has a direct correlation with the number of the Bi_2_Se_3_ nanoplates tilted relative to the substrate surface under the angles of 35°, 54°, and >65°, respectively, which was confirmed by the AFM [35] and SEM [13] characterization.

These XRD peaks are the most intensive for the Bi_2_Se_3_ nanostructures grown on 3 nm (nanoparticle size ~10 nm) thick Au layer (Figure 5b), indicating the highest number of the tilted nanoplates in comparison to the other samples and explaining the highest yield of the Bi_2_Se_3_ nanoribbons per area unit (Figure 3a), as it was reported previously that the tilted nanoplates promote the nanoribbon growth [13]. Presumably, the large number of tilted nanoplates originates from their growth initiated by the VLS growth mechanism with prevailing Bi_2_Se_3_ nucleation mode on the top surface of the Au nanoparticles due to their small diameters [31,32]. Formed on the top of the catalyst nanoparticles, these nanoplates promote further catalyst-free growth of the nanoribbons. At the same time, ~20–30% of the Bi_2_Se_3_ nanoribbons grown on this sample had a Au nanoparticle on its tip, indicating a VLS growth mechanism with the nucleation on the bottom surface of the catalyst, which is expected for the Au catalyst nanoparticles of large diameters. In this case, the width of the nanoribbon is governed by the size of the Au nanoparticle. While typically, the VLS-grown anisotropic nanoribbons are wider than the diameters of the catalyst nanoparticles [36], the size of the nanoparticle limits the maximal widths of the nanoribbons. For example, the widths of the nanoribbons grown on 20 nm Au nanoparticles typically do not exceed 100 nm for the nanoparticles with a diameter ~20 nm [37]. Thus, the contribution of the relatively narrow VLS-grown nanoribbons results in an indicative decrease of their mean widths for the sample grown on 1.5 nm and 3 nm (nanoparticle size ~8–10 nm) thin Au layers in comparison with the nanoribbons grown by catalyst-free method (Figure 3g).

The sample grown on the 13 nm thin Au layer (mean Au nanoparticle diameter ~85 nm) had less intensive XRD peaks related to the tilted nanoplates (Figure 5b) in comparison with the sample synthesized on a 3 nm thin Au layer, which may explain the decrease of the areal number density of the nanoribbons grown on 13 nm thin Au layer (Figure 3a). At the same time, significantly higher in comparison with the samples synthesized on 3 nm Au layer proportion of the nanoribbons presented a Au nanoparticle on its tip (~60–70%) (Figure 4b). This indicates VLS [5] or even VSS [31] growth mechanism with Bi_2_Se_3_ nucleation and growth occurring on the bottom surface of the nanoparticle, which is expected for the Au catalyst nanoparticles with diameters above 20 nm at the growth temperature used in the experiments (~450 °C) [32]. Larger Au nanoparticle diameters result in larger widths of the grown nanoribbons due to the higher throughput of the source vapor. Additionally, the relatively large and rarely located Au nanoparticles formed by the dewetting of thicker than 5 nm (Figure 2) Au layers allow growth of the planar (with c-axis oriented perpendicularly to the substrate surface) Bi_2_Se_3_ nanoplates between the Au nanoparticles, which is proved by the presence of dominating (003) diffraction peak at 2Θ 9.26° in the XRD pattern of this sample (Figure 5). Some of the planar nanoplates become seeds for the catalyst-free growth of the Bi_2_Se_3_ nanoribbons, which are normally wider in comparison with the catalyst-grown (Figure 4c) [29]. The total contribution of the widths of the VLS-grown and catalyst-free grown nanoribbons to the statistics results in a slight increase of the mean widths of the nanoribbons with the increase of the thickness of the initial Au layer above 3 nm and diameters of the Au nanoparticles above 20 nm (Figure 3g,h).

The statistical results on the dependence of the mean length, thickness, and width of the Bi_2_Se_3_ nanoribbons clearly show that the required geometry of the nanoribbons can be obtained by choice of the initial thickness of the Au layer or by choice of diameter of catalyst nanoparticles; however, practical applications of the nanoribbons often require their transfer to other surfaces. Depending on the initial thickness of the Au layer and correspondingly, the diameters of the Au nanoparticles formed by dewetting of these layers, the nanoribbons of certain widths may be selectively transferred by a mechanical pressing process. In this process, only the nanoribbons having weak adhesion to the substrate/nanoplate seed and freestanding above the substrate surface are transferred. The mean width of the mechanically transferred Bi_2_Se_3_ nanoribbons vs. thickness of the initial Au layer showed a minimum for the catalyst-free synthesis and a maximum for the sample synthesized on a 3 nm thin Au layer. The mean widths of the nanoribbons transferred from the samples grown on thicker Au layers showed a tendency for a slight decrease followed by a slight increase, but these deviations were within error limits (Figure 6a).

The width distribution histogram (Figure 6b) showed the differences in statistically prevailing nanoribbon widths in samples synthesized on Au layers of different thicknesses. The largest percentage (~25–35%) of the Bi_2_Se_3_ nanoribbons narrower than 100 nm was transferred from the samples synthesized on Au layers with thicknesses 3 nm and 9 nm. The percentage of Bi_2_Se_3_ nanoribbons wider than 300 nm was maximal (~20%) for the sample synthesized on a 1.5 nm thin Au layer and gradually decreased with the increase of the Au layer thickness. In turn, ~90% of the nanoribbons transferred from the sample grown on a 13 nm thin Au layer had widths between 100 and 300 nm. These findings are important for the applications employing blind transfer of the nanoribbons due to the high probability that the nanoribbons of the required geometry will be transferred to the desired locations.

### 3.3. Carrier Transport Properties

Due to their high yield, Bi_2_Se_3_ nanoribbons from the synthesis with an initial Au layer thickness of 3 nm were selected for electron transport property measurements. In total, four nanoribbons of different thicknesses (32–120 nm) were measured. A SEM image of one of the nanoribbon devices is shown in Figure 7a. Temperature dependence of the sheet resistance for all the measured nanoribbons is plotted in Figure 7b ($R_{xxSheet}=R_{xx}w/L$, where $R_{xx}$ is the resistance measured with a four-probe configuration, L is the length and w is the width of a nanoribbon). Its nearly linear behavior indicates metallic conduction frequently observed in Bi_2_Se_3_ [38,39,40]. 

Magnetoresistance R_xx_(B) dependence at a temperature of 2 K shows oscillations at the magnetic field above ~4 T (Figure 7c, nanoribbon NW3, t = 120 nm). These are the Shubnikov–de Haas oscillations (SdHO). Residual resistance $\Delta R_{xx}$ obtained by removal of polynomial background is not strictly periodic in 1/B (Figure 7d), and Fourier transform gives three frequencies, all below 100 T (see in the inset of Figure 7c). The multiple-frequency SdHO (3 frequencies, all below 100 T) were also recorded for the three other nanoribbons of smaller thicknesses (32, 45, and 56 nm). The multi-frequency SdHO pattern has previously been reported for Bi_2_Se_3_ nanoribbons with thicknesses above ~30 nm [13,15,40,41]. Here, measured nanoribbons are of relatively large thicknesses, and the complex pattern of the SdHO originates from the coexistence of the 3D bulk carriers, the 2D topological surface states, and (or) trivial 2D states [13,40,41].

The temperature dependence of the SdHO (Figure 7d) was measured to determine the cyclotron mass m_c_. The values of m_c_ at a given magnetic field were estimated by fitting the oscillation amplitude $\Delta R_{xx}$ temperature dependence with the Lifshitz–Kosevich (LK) theory [42]. The $\Delta R_{xx}$(T) dependence at B = 8 T with the corresponding LK fit is plotted in the inset of Figure 7e, and determined masses are shown in Figure 7e. The calculated m_c_ varies from ~0.105 to 0.147 m_e_, where m_e_ is the electron mass. The previously reported value of 0.13 m_e_ determined by the ARPES of Bi_2_Se_3_ bulk single crystal [39] fits well within this range, indicating the high quality of the Bi_2_Se_3_ nanoribbons synthesized in this work.

Figure 7f summarizes determined $R_{xxSh.}$ for all four measured Bi_2_Se_3_ nanoribbons. These data are comparable to the sheet resistances of Bi_2_Se_3_ nanoribbons synthesized using the: (1) catalyst-free PVD on a glass substrate, (2) catalyst-free PVD on AAO substrate, and (3) Au-PVD approach. The nanoribbons grown using the 3 nm-Au-PVD approach have an $R_{xxSh.}$ approximately 2–3 times higher. This implies lower carrier density and indicates that the chemical potential in these nanoribbons could be tuned more efficiently by gating techniques, which is important in accessing the transport via the surface Dirac states.

## 4. Conclusions

Physical vapor deposition of Bi_2_Se_3_ nanoribbons on dewetted Au layers allows tuning Bi_2_Se_3_ nanoribbon morphology and yield through the initial Au layer thickness. The initial Au layer thickness impacts the size and number density of Au nanoparticles and, consequently, the yield of synthesized Bi_2_Se_3_ nanoribbons—the highest nanoribbon number density is achieved at an initial Au layer thickness of 3 nm. There are more tilted Bi_2_Se_3_ nanostructures synthesized on dewetted Au, compared to catalyst-free syntheses. A larger amount of tilted Bi_2_Se_3_ nanostructures corresponds to a larger nanoribbon number density. Changes in nanoribbon geometry suggest that the dominating Bi_2_Se_3_ nanoribbon growth mechanism possibly depends on the diameter and number density of Au nanoparticles. For the initial Au layer thicknesses below 5 nm (average size of Au nanoparticles ~8–10 nm), the catalyst-free Bi_2_Se_3_ nanoribbon growth from the tilted relative to the substrate surface Bi_2_Se_3_ nanoplate seeds dominates. For the initial Au layers of thicknesses above 5 nm (average sizes of Au nanoparticles 20–150 nm), the vapor–liquid–solid mechanism with nucleation at the bottom surface of the Au catalyst nanoparticle becomes dominant.

Different frequencies extracted from observed Shubnikov–de Haas oscillations in the synthesized nanoribbons indicate the presence of 3D bulk carriers, as well as 2D topological surface states and (or) trivial 2D states. Determined values of the sheet resistance are about two times higher compared to the ones reported for catalyst-free synthesized nanoribbons, the potential for improving tunability of chemical potential via electrostatic gating and accessing the transport via topological surface states. This confirms that the use of thermally dewetted Au catalyst layers for the high-yield synthesis of Bi_2_Se_3_ nanoribbons with tuned morphology does not result in degradation of transport properties of the nanoribbons. The possibility of adjusting synthesis parameters in order to tune the yield and morphology of synthesized Bi_2_Se_3_ nanoribbons and good transport properties makes them great candidates for applications in nanoelectromechanical devices, where the geometry of the active element determines the operational parameters of the device. The mechanical transfer of the Bi_2_Se_3_ nanoribbons grown on Au layers of different initial thicknesses and, respectively, on Au nanoparticles of different diameters is found to be selective to the nanoribbon geometry. This allows transferring to the other substrates the nanoribbons of required geometry with an efficiency of up to 90%.

## Figures and Tables

**Figure 1 nanomaterials-11-02020-f001:**
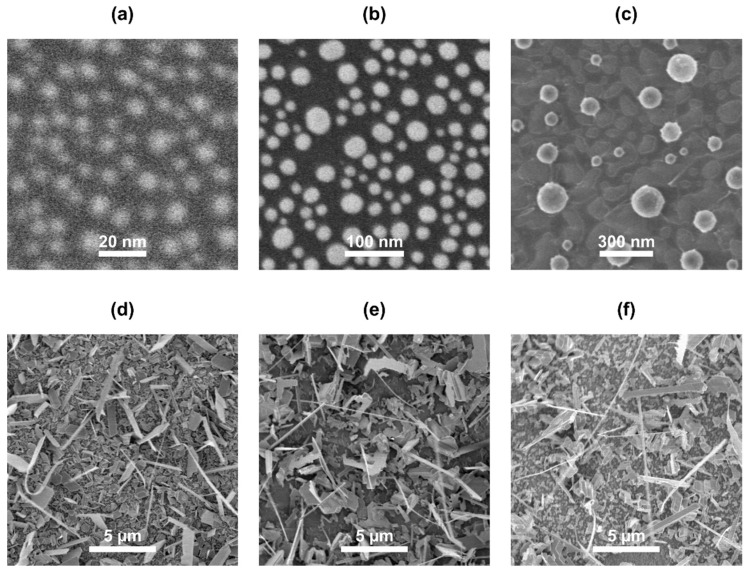
SEM images: (**a**–**c**) Thermally dewetted Au layers with various initial layer thicknesses: (**a**) 3 nm; (**b**) 9 nm; (**c**) 13 nm; (**d**–**f**) Bi_2_Se_3_ nanostructures synthesized on dewetted Au layers with various initial layer thicknesses: (**d**) 3 nm; (**e**) 9 nm; (**f**) 13 nm.

**Figure 2 nanomaterials-11-02020-f002:**
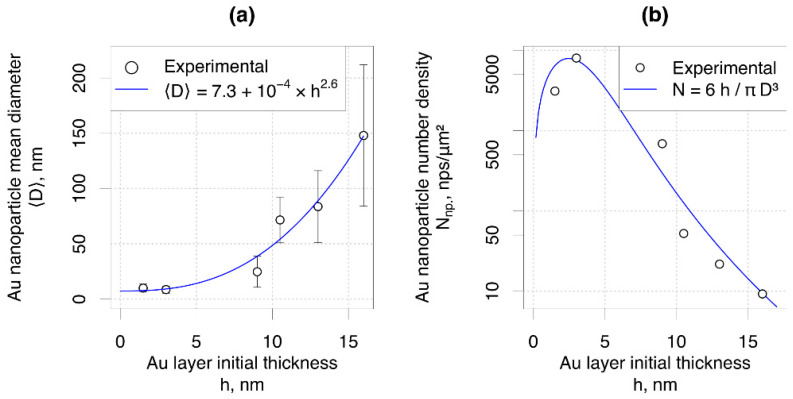
(**a**) Mean Au nanoparticle diameter, the blue line is a fitted power function; (**b**) Au nanoparticle number density (the blue line is the expected number density).

**Figure 3 nanomaterials-11-02020-f003:**
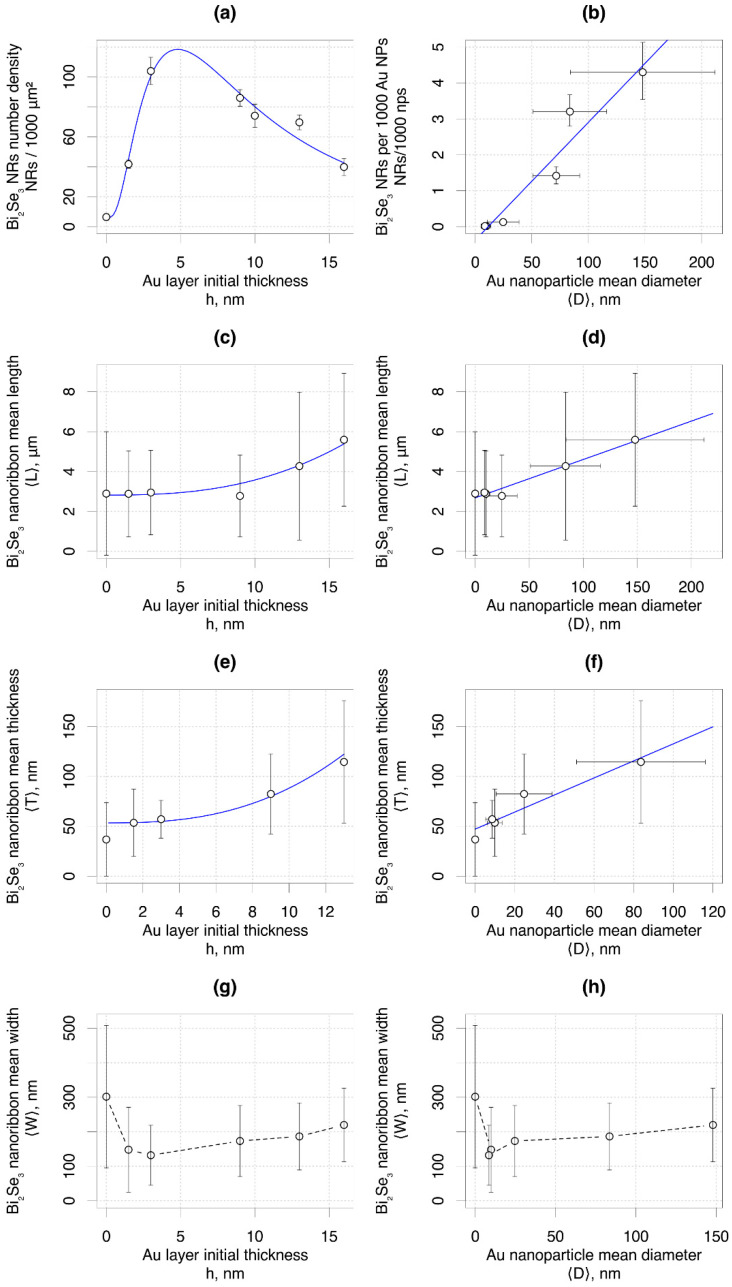
Various Bi_2_Se_3_ nanoribbon parameters as a function of initial Au layer thickness: (**a**,**c**,**e**,**g**) and Au nanoparticle mean diameter (**b**,**d**,**f**,**h**). The blue lines are guides to the eye.

**Figure 4 nanomaterials-11-02020-f004:**
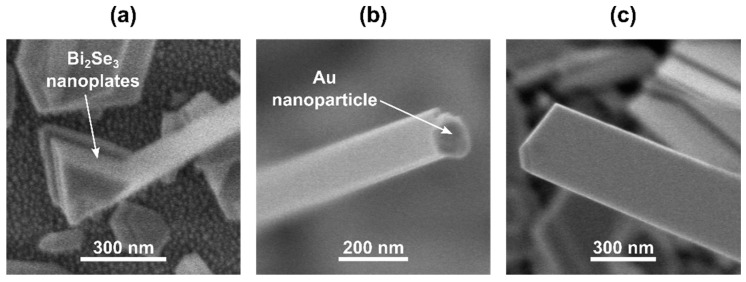
SEM images of synthesized Bi_2_Se_3_ nanostructures: (**a**) Nanoplates at the base of a nanoribbon; (**b**) nanoribbon with a Au nanoparticle at the tip; (**c**) nanoribbon with no Au nanoparticle droplet at the tip.

**Figure 5 nanomaterials-11-02020-f005:**
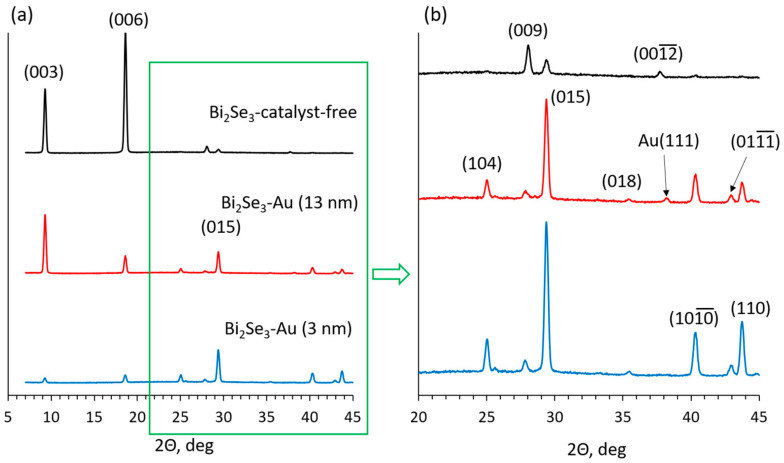
(**a**) XRD spectra of Bi_2_Se_3_ nanostructures grown on glass by the catalyst-free method (black curve), 13 nm thick dewetted Au layer (red curve), and 3 nm thick dewetted Au layer (blue curve); (**b**) closer look at the XRD diffraction peaks corresponding to the tilted Bi_2_Se_3_ nanostructures (ref. card No. 96-901-1966).

**Figure 6 nanomaterials-11-02020-f006:**
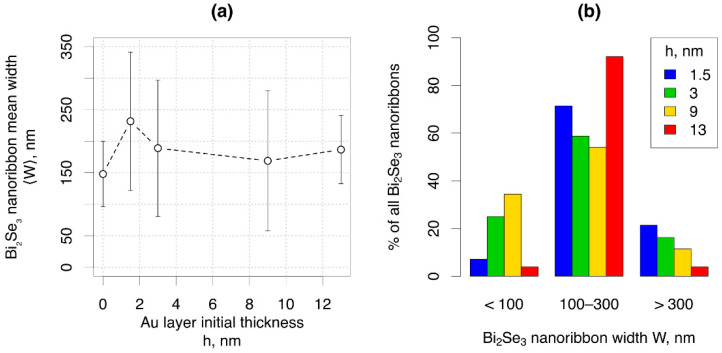
(**a**) Width distribution of Bi_2_Se_3_ nanoribbons mechanically transferred from the dewetted Au layers of various thicknesses; (**b**) width distribution histogram.

**Figure 7 nanomaterials-11-02020-f007:**
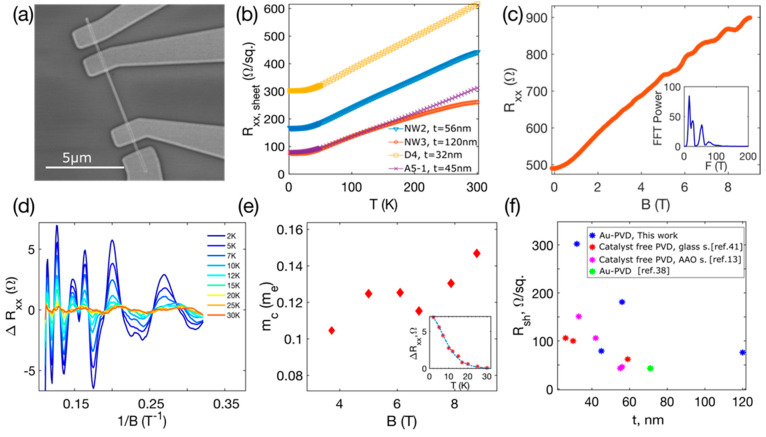
Magnetotransport in 3 nm Au-PVD Bi_2_Se_3_ nanoribbons: (**a**) SEM image of Bi_2_Se_3_ nanoribbon device used in transport measurements (nanoribbon A5-1); (**b**) sheet resistance $R_{xxSheet}$
($=R_{xx}w/L$) as a function of temperature. (**c**) Magnetoresistance $R_{xx}$ of the nanoribbon NW3 measured at 2 K. Inset: FFT spectra of Shubnikov–de Haas oscillations. (**d**) Shubnikov–de Haas oscillations with the subtracted background of the nanoribbon NW3, measured at temperatures 2–30 K. (**e**) Cyclotron mass versus magnetic field. Inset: oscillation amplitude as a function of temperature, dashed blue line is the LK fit yielding $m_{c}=0.130m_{e}$ and the data correspond to the 2nd peak from the left in the $\Delta R_{xx}$(1/B) plot. (**f**) Sheet resistance versus nanoribbon thickness for Bi_2_Se_3_ nanoribbons synthesized using different approaches: catalyst-free PVD on a glass substrate, data from [41], catalyst-free PVD on anodized alumina (AAO), data from [13], Au-PVD, data from [38].

## Data Availability

The data presented in this study are available on request from the corresponding author.

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
