# Peer review of "High-Yield Growth and Tunable Morphology of Bi2Se3 Nanoribbons Synthesized on Thermally Dewetted Au"

_nanomaterials, 2021, doi:10.3390/nano11082020_

Round 1

Reviewer 1 Report

In this work authors systematically studied the tunning of Bi2Se3 nanoribbon yield and morphologies of the structures fabricated by Au-catalyst particles assisted by physical vapor deposition. The work is interesting and appropriate for the MDPI journal of nanomaterials. There are minor corrections or comments below that should be addressed before publication.

Comments:

  1. Figures should be updated with the high-resolution images i.e., figure 1, Figure 2, Figure 3, Figure 4, Figure 5
  2. References should thoroughly revise, specifically the journal abbreviations.

Reviewer 2 Report

With dewetted Au layers, the authors synthesized Be2Si3 nanoribbons by VLS and VSS growth, and investigated the relationship between Au thickness and nanoribbons morphology. The size effect of Au nanoparticles was also mentioned. Some questions are as follows.

Q1. The authors compared the relationship between the various properties of the nanoribbon and the thickness of the gold layer in Figure 3. The relationship between the various properties of the nanoribbon and the size of the gold nanoparticle was also discussed. However, there are no SEM images to support those numerical distributions. It is necessary to add the SEM images of the samples.

Q2. In the manuscript, the mean length and thickness of the synthesized Bi2Se3 nanoribbons had positive correlation with the increasing Au layer thickness, while the width did not. Why is that? Could the width of nanoribbons not be changed? This should be explained in details.

Q3. It was mentioned that if the angle between the Au nanoparticle and Bi2Se3 nanowire surface is constant, the nanoribbon thickness is directly proportional to nanoparticle diameter in line 164 of the manuscript. Please explain this in details.

Q4. In Figure 5 a), what is the nanoflake at the base of a nanoribbon? Where is the Au nanoparticle in Figure 5 b)?

Q5. The JCPDS card should be added for the XRD pattern. Also, it was mentioned that the intensity increase of the diffraction peaks was related to other Bi2Se3 crystallographic planes (104), (015), (1010), (110). It is essential to confirm the lattice structure and the growth direction with another technique. TEM analysis is highly suggested.

Q6. The authors should describe the difference between VLS and VSS methods. Additionally, which method dominates from 3nm layer substrate to 13nm layer substrate?  

Q7. According to reference 22, the lack of kinetic energy is attributed to low temperature. This work looks like a different case. Please explain this clearly.

Q8. It is mentioned that four Bi2Se3 nanoribbons synthesized with 3 nm thick Au layer were selected for electron transport property measurements. Why the authors chose only NW2 and NW3? Why is the resistance value so different in Figure 7 b)?

Q9. The last Figure should be Figure 7 instead of Figure 6. Mistake in the figure caption.

Reviewer 3 Report

The authors demonstrate the high yield growth and tunable morphology of Bi2Se3 nanoribbon using Au nanostructures as a catalyst material. The results are interesting and well-studied; however, this manuscript needs a major revision in order to publish in Nanomaterials.

Comments:

  1. I suggest revising the abstract. Authors should focus on the important findings of the work and present this in a systematic way.
  2. In the Introduction section: What are the applications of Bi2Se3? The author should add important applications of Bi2Se3 nanoribbon.
  3. In the Introduction section: a more discussion of Au catalyst is needed.
  4. Introduction section: Ref. [17] also used Au catalyst. The author should clearly discuss the distinction between the two papers and the novelty of this manuscript.
  5. In Figures 1 and 5: What type of images are presented? SEM? The author should mention this in the caption.
  6. In Figure 4: The author nicely presented the Bi2Se3 nanoribbon parameter as a function of Au nanoparticle diameter. I suggest Author add related SEM images of Au nanoparticles and Bi2Se3 nanoribbon.
  7. In Figure 5, only small-scale SEM images are presented. It would be better to show the large-scale SEM image of Bi2Se3 nanoribbon.
  8. There are two Figures 6 in the manuscript. Please revised it.
  9. References: The author cited all the references published before 2020. I suggest the author cite at least 5-6 recent publications.
  10. Overall: The author should improve the sentence structures, correct style, and grammar errors.

Round 2

Reviewer 2 Report

The authors have properly replied to most of the comments except the following one:

Comment: From response 2 of the previous review report, the authors mentioned that the independence of the mean width of the nanoribbons on the thickness of Au catalyst layer is most likely due to the significant scatter of the measured actual widths of the nanoribbons. They also mentioned that the widths of the nanoribbons depend on the prevailing nucleation and growth mode and may be governed either by the size of Au nanoparticle or by the nanoplate-seed.

However, it is found that the mean diameter of the Au nanoparticles has positive correlation with the increasing Au layer thickness in Figure 2.a) of the manuscript. There is contradiction between the two conclusions; this should be explained detailedly and carefully.

Reviewer 3 Report

The revised version of the manuscript is now acceptable.

Comment: It would be better to check the manuscript carefully once again to eliminate the minor spelling errors and grammar.
